# OpenReview forum: "Conflict-Aware Adaptive Alignment for LLM Hallucination Mitigation"
_ICML.cc/2026/Conference — ICML 2026 regular_

### Official Review · Reviewer_yuAA · 2026-03-08

**Soundness:** 3
**Presentation:** 3
**Significance:** 3
**Originality:** 3
**Overall Recommendation:** 4
**Confidence:** 4

**Summary:**

This paper views LLM hallucination not simply as a knowledge deficiency problem, but as an alignment conflict between truthfulness and helpfulness. To address this issue, it proposes CAMP, which adjusts helpfulness only when such goal conflicts arise, and modulates the strength of this adjustment differently for each sample.

**Compliance With Llm Reviewing Policy:**

Affirmed.

**Key Questions For Authors:**

1. The main training comparisons are conducted on Llama-3.1-8B-Instruct. Do the benefits of CAMP also hold across other base models?

2. How sensitive are the reported results to the data construction choices?

**Limitations:**

Please refer to the weaknesses above.

**Strengths And Weaknesses:**

Strengths

- The paper reframes hallucination mitigation as a problem of partial conflict between truthfulness and helpfulness.

- Rather than fully replacing DPO, the method introduces a pair-dependent margin and applies regularization only when conflict is present. This design is intuitive and convincing: it preserves strong truthfulness signals when the objectives are synergistic, while relaxing optimization only for conflict cases.

Weaknesses

- The experiments focus on a single base model and a relatively clean preference-pair construction setting, so generalization to other model scales, other preference datasets, or noisier annotation environments is not yet sufficiently established.

- The paper does not provide enough analysis of how dataset construction affects the results. In particular, it is unclear how much the use of 20k pairs and the filtering of pairs with a score gap of at least 1 influence the observed conflict ratio and the effectiveness of margin modulation.

---

> ### Author Rebuttal · Authors · 2026-03-31
>
> We thank the reviewer for the valuable and helpful feedback. We respond to the main concerns below:
>
> **W1 & Q1 (generalization to other model scales, datasets, annotation environments):**
> * Due to the high computational cost of preference alignment training, 7B-8B (or smaller) models are commonly used in prior work. In addition, existing preference datasets (e.g., HH-RLHF) typically provide responses annotated for a single objective, rather than multi-objective scores on each response. This limitation in current datasets also motivates our focus on modeling interactions between objectives, which is less explored in existing settings.
> * Following the reviewer’s suggestion, we conduct additional experiments on a different backbone, Qwen2.5-7B-Instruct, evaluated on TruthfulQA. The results show consistent improvements over the strongest baseline (MODPO) across all metrics, demonstrating that the benefits of CAMP generalize beyond a single base model.
> | |Helpful↑|Truthful↑|Accuracy↑|Hallu↓|
> |---|---|---|---|---|
> |MODPO|4.13|4.55|88.62|7.22|
> |Ours|**4.15**|**4.62**|**90.45**|**5.88**|
> * We further explore robustness under an alternative annotation setting by replacing truthfulness with another honesty score in UltraFeedback. The results again show consistent performance gains, suggesting that our method remains effective under different supervision signals.
> | |Helpful↑|Truthful↑|Accuracy↑|Hallu↓|
> |---|---|---|---|---|
> |MODPO|3.87|4.22|80.17|12.48|
> |Ours|**3.90**|**4.34**|**83.72**|**9.79**|
> * Moreover, we acknowledge that generalization to larger-scale models and more diverse annotation environments remains an open question. We will add this as a limitation and include these additional analyses in the revised paper to further strengthen the empirical evidence of our approach.
>
> **W2 & Q2 (sensitivity to data construction choices):**
> * We first analyze the impact of data construction on the observed conflict ratio by varying both the number of training pairs and the score-difference filtering. We find that the conflict ratio is not sensitive to the number of pairs, remaining highly consistent across 15k, 20k, and 25k settings. In contrast, as the required truthfulness score gap increases, the proportion of conflict decreases substantially. This suggests that larger factual improvements are typically aligned with helpfulness, while conflicts primarily arise in more marginal cases.
> | | |15k|20k|25k|
> |---|---|---|---|---|
> | |1|16.26|16.34|16.22|
> |Score Gap|2|13.71|13.84|13.65|
> | |3|6.41|6.74|6.55|
> * Regarding the score gap filtering ($\ge 1$), we note that this is typically standard practice in preference learning rather than a tunable hyperparameter. Specifically, a gap of 0 indicates no difference between responses, and thus cannot provide a meaningful preference training signal. Using larger thresholds ($> 1$) would manually filter out a substantial portion of training data and reduce data efficiency, and is therefore typically not adopted unless there is a specific need.
> * We examine the impact on model performance by varying the training size (15k/20k/25k preference pairs) on TruthfulQA. The results show consistent improvements over the best baseline MODPO across all settings, with performance stabilizing as the dataset size increases. This indicates an empirically convergent setting and suggests that our method is not sensitive to the specific choice of training size. We will include these additional analyses in the revised paper to further strengthen the discussion on sensitivity to data construction choices.
> | |15k| |20k| |25k| |
> |---|---|---|---|---|---|---|
> | |Help↑|Truth↑|Help↑|Truth↑|Help↑|Truth↑|
> |MODPO|3.93|4.25|3.91|4.30|3.90|4.25|
> |Ours|**3.96**|**4.40**|**3.97**|**4.41**|**3.95**|**4.43**|
>
>   | |15k| |20k| |25k| |
>   |---|---|---|---|---|---|---|
>   | |Acc↑|Hallu↓|Acc↑|Hallu↓|Acc↑|Hallu↓|
>   |MODPO|80.54|13.46|82.38|11.63|81.76|12.36|
>   |Ours|**85.92**|**9.30**|**85.92**|**9.79**|**86.17**|**9.79**|

---

> > ### Author Rebuttal · Reviewer_yuAA · 2026-04-03
> >
> > Thank you for the detailed response. While most of the concerns raised in my previous comment have been addressed, the following questions remain:
> >
> > - W1: It is unfortunate that the experiments were only conducted on TruthfulQA.
> >
> > - W2: I am also curious about the performance in environments where data is much more limited.

---

> > > ### Author Response · Authors · 2026-04-07
> > >
> > > We appreciate the follow-up and are glad that most concerns have been addressed. We address the remaining questions below.
> > >
> > > **W1 (evaluation on additional model backbone and alternative honesty score):**
> > > * We agree that evaluating only on TruthfulQA is limited. Our initial additional experiment focused on this benchmark due to the rebuttal timeline. Following the reviewer’s suggestion, we have now extended the evaluation to all three benchmarks (UltraFeedback, TruthfulQA, HaluEval), using both the different backbone (Qwen2.5-7B-Instruct) and the alternative honesty score as supervision. CAMP consistently outperforms the strongest baseline, MODPO, across these settings. These results demonstrate that our method generalizes across backbones and under different annotation environments. We will include the comprehensive evaluation results in our paper.
> > >
> > > * Qwen2.5-7B-Instruct:
> > > | |Ultra| |TruthQA| |HaluEval| |
> > > |---|---|---|---|---|---|---|
> > > | |Helpful↑|Truthful↑|Helpful↑|Truthful↑|Helpful↑|Truthful↑|
> > > |MODPO|3.89|4.45|4.13|4.55|4.73|4.79|
> > > |Ours|**3.91**|**4.48**|**4.15**|**4.62**|**4.75**|**4.80**|
> > >
> > >   | |Ultra| |TruthQA| |HaluEval| |
> > >   |---|---|---|---|---|---|---|
> > >   | |Accuracy↑|Hallu.↓|Accuracy↑|Hallu.↓|Accuracy↑|Hallu.↓|
> > >   |MODPO|86.32|9.63|88.62|7.22|94.40|4.40|
> > >   |Ours|**87.27**|**9.31**|**90.45**|**5.88**|**95.05**|**3.95**|
> > >
> > > * Alternative Honesty Score:
> > > | |Ultra| |TruthQA| |HaluEval| |
> > > |---|---|---|---|---|---|---|
> > > | |Helpful↑|Truthful↑|Helpful↑|Truthful↑|Helpful↑|Truthful↑|
> > > |MODPO|3.80|4.37|3.87|4.22|4.64|4.77|
> > > |Ours|**3.82**|**4.40**|**3.90**|**4.34**|**4.66**|**4.78**|
> > >
> > >   | |Ultra| |TruthQA| |HaluEval| |
> > >   |---|---|---|---|---|---|---|
> > >   | |Accuracy↑|Hallu.↓|Accuracy↑|Hallu.↓|Accuracy↑|Hallu.↓|
> > >   |MODPO|83.76|11.27|80.17|12.48|94.30|3.85|
> > >   |Ours|**84.86**|**9.91**|**83.72**|**9.79**|**95.15**|**3.40**|
> > >
> > > **W2 (performance on more limited data size):**
> > > * We appreciate the reviewer’s suggestion. We further evaluate under much more limited data regimes (2.5k-10k pairs). Notably, preference alignment typically requires larger datasets; settings below 10k are rarely used for stable performance, and here mainly serve to analyze early-stage training dynamics rather than final performance.
> > > * Results show that MODPO achieves slight early helpfulness gains but quickly plateaus and fluctuates without consistent improvement. In contrast, our method exhibits more stable scaling behavior, with consistent improvements in helpfulness, truthfulness, accuracy, and hallucination reduction as data increases from 2.5k to 10k/15k. It then converges to a clearly stronger performance regime at larger scales (15k-25k). We will include this analysis to further demonstrate the learning dynamics under varying training sizes, showing improved and stable scaling behavior of our method.
> > > | |2.5k| |5k| |7.5k| |10k| |15k| |20k| |25k| |
> > > |---|---|---|---|---|---|---|---|---|---|---|---|---|---|---|
> > > | |Help↑|Truth↑|Help↑|Truth↑|Help↑|Truth↑|Help↑|Truth↑|Help↑|Truth↑|Help↑|Truth↑|Help↑|Truth↑|
> > > |MODPO|3.89|4.17|3.91|4.21|3.93|4.17|3.91|4.21|3.93|4.25|3.91|4.30|3.90|4.25|
> > > |Ours|3.83|4.15|3.86|**4.23**|**3.94**|**4.29**|**3.97**|**4.31**|**3.96**|**4.40**|**3.97**|**4.41**|**3.95**|**4.43**|
> > >
> > >   | |2.5k| |5k| |7.5k| |10k| |15k| |20k| |25k| |
> > >   |---|---|---|---|---|---|---|---|---|---|---|---|---|---|---|
> > >   | |Acc.↑|Hallu.↓|Acc.↑|Hallu.↓|Acc.↑|Hallu.↓|Acc.↑|Hallu.↓|Acc.↑|Hallu.↓|Acc.↑|Hallu.↓|Acc.↑|Hallu.↓|
> > >   |MODPO|78.70|12.48|79.68|12.00|77.85|14.08|79.44|13.95|80.54|13.46|82.38|11.63|81.76|12.36|
> > >   |Ours|77.85|14.44|**80.17**|**11.87**|**82.99**|**10.40**|**82.37**|**10.40**|**85.92**|**9.30**|**85.92**|**9.79**|**86.17**|**9.79**|

---

### Official Review · Reviewer_Nsyi · 2026-03-09

**Soundness:** 3
**Presentation:** 3
**Significance:** 3
**Originality:** 3
**Overall Recommendation:** 4
**Confidence:** 4

**Summary:**

The paper argues that truthfulness and helpfulness are not always conflicting objectives when mitigating hallucinations in LLMs and proposes CAMP to model when these conflicts actually arise and to adaptively regulate optimization strengths, unlike in typical RLHF settings. CAMP is tested on benchmarks to  demonstrate its hallucination mitigation capabilities.

**Compliance With Llm Reviewing Policy:**

Affirmed.

**Final Justification:**

The main points raised in the original still hold. The statistical significance was useful. The performance improvement on T*I, which is the more useful metric on TFQA seems to be only marginally better than the other methods. On balance, I will keep my score.

**Key Questions For Authors:**

1. It is not explicitly stated, but is the expectation that the value of $\lamda$ supposed to be between 0 and 1?
2. The performance improvement of the proposed method over the MODPO/CPO is much smaller than the performance  improvement  posted by the MODPO/CPO over the baseline methods. Does this indicate that there may not be much to be gained from tweaking at the training level, atleast for the tasks considers?
3. Can the authors provide some statistical significance of their metrics (especially in light of the above point)
4. Could the authors provide some more information on the human annotators? How many were used per sample? Where were they recruited from? I more than one human annotators were used per sample, how did their inter-annotator agreement fare compared to that between humans and GPT?

**Limitations:**

Notes on the optimization of the hyperparameter , lamda, are appreciated, but it would be good to also see some discussions on the potential limitations of the method on other tasks like summarization, etc. since this has not been tried out on those tasks.

**Strengths And Weaknesses:**

Strengths of the paper:
+ The empirical insight using the TruthfulQA and UltraFeedback datasets is useful.
+ Modulating the regulation of helpfulness only when the objective conflicts with the truthfulness objective, is novel and makes sense in light of the observation mentioned above.
Weaknesses
- I am unsure that the claim, "most work in this area treat truthfulness and helpfulness as universally conflicting" is actually true. It is possibly true in the training based solutions community, but it is not always treated that way in the decoder based solutions.
- It is also untrue that inference time measures don't address the "true cause of hallucinations", since hallucinations can be caused because the LLM does not know the answer (the case that the present method is seeking to address) and because it ignores the correct answers sometimes even when it has that answer as one of the options at inference time. However, the authors do seem to be aware that these methods can be complementary
-  The performance improvement of the proposed method over the MODPO/CPO is very small compared to the performance improvement posted by MODPO/CPO on the baseline.
- Although the authors mention that this method could be used for  other tasks, they only present results for QA tasks. Perhaps they should consider narrowing the scope of their contribution by suitably modifying the title

---

> ### Author Rebuttal · Authors · 2026-03-31
>
> We thank the reviewer for the thoughtful and detailed feedback. We address the main concerns below:
>
> **W1 (expression of “universally conflicting”):**
> * We agree it is inaccurate to state that most work treats truthfulness and helpfulness as universally conflicting. In our paper, we instead state that existing alignment methods often treat these objectives as either universally collaborative or conflicting. We also acknowledge that this characterization mainly applies to training-based methods. In contrast, inference-time approaches (e.g., decoding, retrieval) can adjust the balance on a per-sample basis. We will revise the wording to clarify this scope and avoid confusion.
>
> **W2 (expression of "true cause of hallucinations"):**
> * We agree that hallucinations can arise from multiple sources, including lack of knowledge and inference-time failures. We will revise our motivation to avoid overclaiming that alignment training addresses the “true cause”. Our goal is not to compete with or replace inference-time approaches, but to study the role of alignment during training. We view training and inference as complementary stages, and our method can be naturally combined with inference-time techniques.
>
> **W3 & Q2 & Q3 (performance improvement relatively small & statistical significance):**
> * We first clarify that prior methods such as MODPO and CPO focus on general multi-objective optimization and typically show gains over relatively vanilla baselines (e.g., SFT/DPO/RLHF). In contrast, our goal is to address the specific truthfulness–helpfulness trade-off in hallucination mitigation, and we compare against strong alignment baselines tailored to this setting. These general methods are also evaluated on different tasks, datasets, and metrics, making improvement magnitudes not directly comparable. In our experiments (Tables 1 and 2), MODPO and CPO do not consistently improve over DPO trained on overall scores for hallucination-focused metrics, suggesting that uniform trade-offs may be less effective in this problem and motivating our conflict-aware design.
> * Following the suggestion, we conduct statistical significance tests to assess robustness. We use paired t-tests on per-sample evaluation scores between our method and each baseline on TruthfulQA, accounting for paired variability across samples. Our method shows consistent, statistically significant improvements across most comparisons, including strong multi-objective baselines. In particular, truthfulness gains are especially strong, with most cases achieving $p<0.001$. These results confirm that our improvements over methods such as MODPO and CPO are robust and statistically meaningful.
> | | |Help| |Truth| |
> |---|---|---|---|---|---|
> | | |t|p|t|p|
> |Ours vs.|SFT|7.119|**0.000**|7.530|**0.000**|
> | |DPO|1.683|0.093|3.751|**0.000**|
> | |AlignHonesty|3.030|**0.003**|5.534|**0.000**|
> | |FLAME|2.666|**0.008**|4.998|**0.000**|
> | |MODPO|2.354|**0.019**|3.024|**0.003**|
> | |CPO|4.575|**0.000**|7.210|**0.000**|
>
> **W4 & Limitation (evaluation task scope):**
> * We acknowledge that our evaluation is relatively QA-heavy, particularly on hallucination-focused benchmarks (TruthfulQA and HaluEval). At the same time, evaluation on the UltraFeedback test set covers a broader range of tasks, including open-ended generation, dialogue, and instruction completion, providing a more general assessment beyond QA settings. We agree that generalizability to additional tasks (e.g., summarization or tool use) is not yet fully established and requires further study. We will add this as a limitation and an important direction for future work.
>
> **Q1 (lambda value range):**
> * $\lambda$ is a non-negative coefficient ($\lambda \ge 0$) controlling the influence of helpfulness degradation on the margin, and is not restricted to $[0,1]$. Empirically, values in $[0,1]$ are more reasonable, as they allow helpfulness to act as a soft regularization signal without overwhelming the truthfulness-driven margin. In practice, we use smaller values (e.g., $\lambda = 0.25$) to keep truthfulness as the primary objective.
>
> **Q4 (human annotator details):**
> * Our evaluation follows common practice in recent LLM alignment work, where LLM-as-a-judge is used for scalable evaluation due to the high cost of large-scale human annotation, typically complemented by smaller-scale human validation rather than full crowdsourcing (e.g., Amazon Mechanical Turk). For example, “Reducing Tool Hallucination via Reliability Alignment” (ICML 25) evaluates 150 sampled cases without multi-annotator labeling.
> * Following this practice, we conduct human evaluation on 100 samples with 3 annotators per sample, all graduate students with NLP/ML backgrounds recruited internally. We measure agreement by averaging the three annotators’ scores per sample and computing correlation with GPT-based evaluation. This provides a practical and reliable validation of our LLM-based evaluation. We will expand the appendix with additional details.

---

> > ### Author Rebuttal · Reviewer_Nsyi · 2026-04-03
> >
> > Thanks for your response. I still have some questions.
> > 1. For the significance test, which scores did you gather from the competing methods? Could you describe that method in better detail?
> > 2. I don't have an issue with the LLM annotations, my question was more focused on what was the inter-annotator score for two groups (1) within the humans (2) between the humans and the LLM judge. It would also be good to know how human scores were aggregated when (2) is calculated.
> > 3. Authors of TruthfulQA explicitly mention that "no comment" responses are counted as "truthful" responses and can artificially inflate %T. Can you provide T*I scores for your methods and the baselines?

---

> > > ### Author Response · Authors · 2026-04-07
> > >
> > > We thank the reviewer for the detailed follow-up questions and the opportunity to further clarify.
> > >
> > > **1. significance test details:**
> > > * For statistical significance testing, we use the standard paired t-test to compare two sets of continuous scores evaluated on the same set of samples, separately for helpfulness and truthfulness scores in Table 1. Specifically, for each method, we use the per-sample helpfulness scores and per-sample truthfulness scores assigned by the LLM judge across all evaluation samples. For each pairwise comparison of ours vs. a baseline, we compare the corresponding per-sample scores between the two methods over the shared evaluation set. This setup evaluates whether the mean differences underlying the averages in Table 1 are statistically significant across evaluation samples. We report the resulting t-statistics and p-values separately for helpfulness and truthfulness scores.
> > >
> > > **2. inter-annotator scores:**
> > > * In our human evaluation, we collect annotations from three annotators per sample, each providing continuous scores on the same scale as the LLM judge.
> > > For human–human agreement, we compute inter-annotator agreement using pairwise Pearson correlations between annotators’ per-sample scores, averaged across all annotator pairs. This yields an average agreement of 0.72, indicating good consistency among human annotators.
> > > For human–LLM agreement, we first aggregate human scores by averaging the three annotators per sample, and then compute the Pearson correlation between the aggregated human scores and the LLM scores. This results in a correlation of 0.70.
> > > These results show comparable agreement levels within humans and between human consensus and the LLM judge, suggesting that the LLM-based evaluation is well aligned with human judgment.
> > >
> > > **3. Truthfulness × Informativeness scores:**
> > > * This is closely aligned with our objective of jointly modeling truthfulness and helpfulness. In our setting, helpfulness serves as a stricter proxy for informativeness, as it additionally requires relevance and usefulness to the question beyond merely providing information.
> > > * Our original evaluation reports truthfulness and helpfulness separately, and compares methods jointly across both metrics. Following the reviewer’s suggestion, we further integrate them into a single metric. To maintain consistency with our evaluation setup (and avoid introducing an additional informativeness metric that may overlap with helpfulness), we use helpfulness as a stricter proxy for informativeness. Specifically, we compute T*I by taking the product of the truthfulness and helpfulness scores for each sample, normalizing it from the range 1-25 (i.e., $1\times1$ to $5\times5$) back to the 1-5 scale, and then averaging across the dataset: $T\*I = \frac{1}{N} \sum_{i=1}^{N} \frac{T_i \cdot I_i}{5}$. Evaluation results show that our method consistently outperforms all baselines under this metric. The improvements are most significant on TruthfulQA, while gains on HaluEval are smaller, likely due to saturation effects across methods in helpfulness. We will include this as an additional metric in the paper.
> > > | Method | UltraFeedback | TruthfulQA | HaluEval |
> > > |---|---|---|---|
> > > | SFT | 3.18 | 3.23 | 4.53 |
> > > | DPO | 3.50 | 3.49 | 4.54 |
> > > | AlignHonesty | 3.37 | 3.36 | 4.47 |
> > > | FLAME | 3.43 | 3.43 | 4.55 |
> > > | MODPO | 3.53 | 3.52 | 4.52 |
> > > | CPO | 3.47 | 3.34 | 4.53 |
> > > | Ours | **3.55** | **3.60** | **4.56** |

---

### Official Review · Reviewer_JSbm · 2026-03-12

**Soundness:** 3
**Presentation:** 3
**Significance:** 3
**Originality:** 2
**Overall Recommendation:** 4
**Confidence:** 3

**Summary:**

This paper addresses a timely and relevant problem: the tendency of alignment-based hallucination mitigation to inadvertently compromise helpfulness through overly conservative outputs. The authors observe that existing methods often treat the truthfulness–helpfulness trade-off as a globally uniform constraint, overlooking sample-specific dynamics. To address this, they propose CAMP, a DPO-based preference optimization framework that introduces a conflict-aware adaptive margin. By categorizing preference pairs as either in 'synergy' or 'conflict' based on helpfulness differentials, CAMP selectively scales the optimization—strengthening truthfulness constraints on non-conflicting pairs while relaxing them where objectives clash. A key empirical insight driving this approach is the relative sparsity of objective conflicts (noted at ~11% in UltraFeedback and ~10.5% in TruthfulQA), which suggests that targeted regulation is indeed more effective than a blanket balancing act.

Experiments are conducted with Llama-3.1-8B-Instruct as the common backbone, trained on 20k preference pairs from UltraFeedback and evaluated on UltraFeedback, TruthfulQA, and HaluEval. Empirically, CAMP generally improves both helpfulness and truthfulness, with especially visible gains on hallucination-related metrics. For instance, on TruthfulQA, CAMP achieves helpfulness 3.97, truthfulness 4.41, accuracy 85.92, and hallucination rate 9.79, outperforming DPO and MODPO consistently. The paper also includes ablations, a sweep over the trade-off hyperparameter λ, and a conflict-distribution analysis to support the claim that it is better to regulate the trade-off only where conflict actually arises.

**Compliance With Llm Reviewing Policy:**

Affirmed.

**Final Justification:**

While not without issues, I believe the paper merits a weak accept.

**Key Questions For Authors:**

Q1. CAMP’s adaptive margin directly depends on the magnitudes of the truthfulness and helpfulness scores. If score calibration is noisy, or if a different judge is used, the meanings of Δtruth and Δhelp may shift substantially. Can the authors provide evidence that the same trends hold under alternative judges or noisier annotation settings?

Q2. The key motivation of the paper is that objective conflict is sparse. Does this observation also hold for other backbones, other model scales, or other alignment datasets beyond UltraFeedback-style preference data? At present, it is not clear how general this phenomenon is.

**Limitations:**

yes

**Strengths And Weaknesses:**

S1. The problem formulation is strong. A key strength of the paper is that it begins by arguing that truthfulness and helpfulness are not uniformly in conflict. The observation that conflict occurs in only around 10% of pairs in UltraFeedback and TruthfulQA directly challenges the coarse assumption behind many existing multi-objective alignment methods. In that sense, the paper does not merely introduce another loss variant; it reframes hallucination mitigation as a sample-wise conflict regulation problem, which is a clear and well-motivated perspective.

S2. The method is simple, but the intuition is clear. CAMP essentially inserts a pair-dependent adaptive margin into the DPO objective: for synergy pairs it uses the truthfulness gap, while for conflict pairs it uses  $\max(\Delta \text{truth} + \lambda \Delta \text{help}, 0)$. This is easy to understand and reasonably elegant: push truthfulness hard when helpfulness is not harmed, and only apply a brake when improving truthfulness would reduce helpfulness. The ablation study is also reasonably convincing, since removing conflict-aware margin activation, removing adaptive strength modulation, or falling back to truthfulness-only DPO all lead to performance drops.

S3. The empirical story is fairly consistent. Across datasets, CAMP improves truthfulness-related scores and lowers hallucination rates, with especially clear gains on hallucination-focused benchmarks such as TruthfulQA and HaluEval. The λ sweep is also useful: moderate λ values improve helpfulness without noticeably hurting truthfulness or accuracy, which matches the paper’s intuition. Similarly, the conflict analysis suggests that conflict is both sparse and typically moderate in strength, which aligns well with the paper’s motivation for adaptive rather than uniform trade-off control.

---
W1. The methodological novelty appears somewhat limited and incremental. CAMP is best understood as a DPO-style preference optimization variant with conflict-aware adaptive margins, rather than a fundamentally new alignment paradigm. While the framing is clear and the design is sensible, the conceptual distance from prior multi-objective preference optimization methods such as MODPO and CPO seems modest. The contribution is meaningful, but the novelty appears to lie more in a clean conflict-aware instantiation than in a substantially new methodological principle.

W2. The evaluation protocol is still fairly judge-dependent. The main results rely heavily on GPT-4.1 as the evaluator, and the human validation appears limited to a 100-sample correlation analysis for truthfulness. That is enough to suggest the evaluation is not arbitrary, but it is not enough to remove concerns altogether. There is no comparable human validation for helpfulness, and the derived accuracy/uncertainty/hallucination categories are still based on thresholding judge-assigned scores. As a result, the direction of the findings seems plausible, but their robustness to evaluator choice is less clear.

W3. The external validity of the results is somewhat limited. The experiments are effectively centered on a single backbone, Llama-3.1-8B-Instruct, with 20k preference pairs and one-epoch training, so it is unclear how well the findings transfer to larger models or other post-training regimes. Moreover, the key claim that objective conflict is sparse is demonstrated primarily on UltraFeedback-style multi-score preference data, and may not hold as cleanly in longer-form generation, reasoning-heavy tasks, or tool-use settings. The current evidence is promising, but the broader generalization claims should be stated more cautiously.

---

> ### Author Rebuttal · Authors · 2026-03-31
>
> We thank the reviewer for the insightful comments and suggestions. We address each point below:
>
> **W1 (method novelty):**
> * We agree that our work builds on DPO-style preference optimization rather than introducing a fundamentally new alignment paradigm. Our goal is to address a key limitation of prior multi-objective methods, which often apply uniform trade-offs. CAMP’s novelty lies in explicitly modeling conflict at the sample level and adaptively adjusting optimization strength. We view this as a targeted, principled extension that improves multi-objective preference optimization for hallucination mitigation.
>
> **W2 (evaluation protocol):**
> * We agree that evaluator robustness is critical. While LLM-based evaluation is not a perfect substitute for human studies, it provides a practical and widely validated proxy, especially given the high cost of large-scale human annotation. Our protocol follows common practice in recent alignment work, where LLM-as-a-judge is widely adopted in general-purpose benchmarks such as MT-Bench and AlpacaEval. We also follow typical human evaluation scales used in prior work (e.g., “Reducing Tool Hallucination via Reliability Alignment,” ICML 25, which samples 150 cases in total).
> * We further conducted human validation for helpfulness and observed a comparable correlation (Pearson $r = 0.72$) between GPT-4.1 and human judgments. This supports the consistency of the LLM judge across both truthfulness and helpfulness. We will add clarification and validation to Section 5.1 and Appendix B.
>
> **W3 (generalization limitation):**
> * Our method is training-based and produces standard model weights, making it directly compatible with post-training pipelines, similar to any pre-trained or tuned models. At the same time, we acknowledge generalization as a limitation and will revise the paper to state more cautiously. We will clarify that the observed sparsity of conflict may not extend to broader settings such as reasoning or tool-use tasks. We use an 8B model following common practice in alignment research, where smaller models are preferred due to high training cost, while larger models are often used for inference. We will clarify this in the limitation section and note that extending to larger models is future work.
> * To further assess robustness, we conduct experiments across varying training sizes (15k/20k/25k pairs). Results on TruthfulQA show consistent improvements over the best baseline, with performance stabilizing across scales, indicating an empirically convergent regime and low sensitivity to training size.
> | |15k| |20k| |25k| |
> |---|---|---|---|---|---|---|
> | |Help↑|Truth↑|Help↑|Truth↑|Help↑|Truth↑|
> |MODPO|3.93|4.25|3.91|4.30|3.90|4.25|
> |Ours|**3.96**|**4.40**|**3.97**|**4.41**|**3.95**|**4.43**|
>
>   | |15k| |20k| |25k| |
>   |---|---|---|---|---|---|---|
>   | |Acc↑|Hallu↓|Acc↑|Hallu↓|Acc↑|Hallu↓|
>   |MODPO|80.54|13.46|82.38|11.63|81.76|12.36|
>   |Ours|**85.92**|**9.30**|**85.92**|**9.79**|**86.17**|**9.79**|
>
> **Q1 (CAMP under alternative judges):**
> * We rely on publicly available UltraFeedback annotations. In addition to truthfulness, the dataset provides an honesty score, capturing a related concept emphasizing subjective honesty over objective factual correctness. The correlation between honesty and truthfulness is 0.67, indicating they are strongly related but not identical, making honesty a meaningful alternative signal. We replace truthfulness with honesty as the primary objective while retaining helpfulness for regularization, and evaluate CAMP against the best baselines on TruthfulQA. Results show consistent gains, demonstrating effectiveness under this alternative signal. This suggests CAMP is not sensitive to a specific judge calibration, but generalizes to related signals with different semantics. We will include this analysis in paper to support robustness.
> | |Help↑|Truth↑|Acc↑|Hallu↓|
> |---|---|---|---|---|
> |MODPO|3.87|4.22|80.17|12.48|
> |Ours|**3.90**|**4.34**|**83.72**|**9.79**|
>
> **Q2 (generality of the conflict sparse observation):**
> * UltraFeedback is a diverse benchmark covering a wide range of tasks (e.g., TruthfulQA, Evol-Instruct, UltraChat, ShareGPT) and responses from models spanning architectures and scales (e.g., GPT-4/3.5, LLaMA 7B-70B, Falcon-40B). This diversity reduces model-specific bias and suggests the observed conflict sparsity is not tied to a single backbone or scale.
> * At the same time, generalizability is limited by the availability of datasets with multi-objective annotations on the same responses. Beyond UltraFeedback, such datasets are scarce. Commonly used datasets (e.g., HH-RLHF) provide annotations for a single objective without joint evaluation. Thus, while our observation is supported by a comprehensive dataset, its generality beyond UltraFeedback-style annotations remains an open question. We will revise the limitation section to highlight the need for broader multi-objective datasets and further validation.

---

> > ### Author Rebuttal · Reviewer_JSbm · 2026-04-03
> >
> > Thanks for the detailed response. I appreciate the clarification on the paper’s methodological contribution, as well as the added human validation and robustness analyses. These help address several of my concerns. At the same time, I still view the evaluation as somewhat judge-dependent, since the main findings are not yet shown to be robust across alternative evaluators or thresholding choices, and the broader generalization claims remain only partially substantiated beyond the current backbone and dataset setting. On balance, I will maintain my score reflecting the positive assessment of the paper.

---

> > > ### Author Response · Authors · 2026-04-07
> > >
> > > We thank the reviewer for the thoughtful feedback and positive assessment of our revisions. We appreciate the recognition of our methodological contributions, human validation, and robustness analyses. We acknowledge the concern regarding potential judge dependency and broader generalization, and will further clarify these aspects in the final version. We also note that LLM-based evaluation is a widely adopted practice in recent alignment work, though we agree it is important to discuss its limitations. We thank the reviewer again for the constructive comments.

---

### Official Review · Reviewer_YBRp · 2026-03-13

**Soundness:** 2
**Presentation:** 3
**Significance:** 3
**Originality:** 2
**Overall Recommendation:** 3
**Confidence:** 4

**Summary:**

This paper addresses the "alignment tax" inherent in mitigating large language model (LLM) hallucinations. Historically, aggressively aligning a model to be truthful (e.g., teaching it to refuse answering when it lacks knowledge) invariably degrades its overall helpfulness, leading to an over-cautious model that refuses perfectly safe and answerable prompts.

The authors propose "Conflict-Aware Adaptive Alignment," built on the hypothesis that truthfulness and helpfulness are actually aligned for the vast majority of training data. They argue that these two objectives actively conflict in only a small, identifiable subset of samples. Instead of treating the truthfulness-helpfulness trade-off as a global parameter during preference optimization (like standard DPO or PPO), the authors introduce a method to isolate these high-conflict samples. They then apply an adaptive alignment strategy that dynamically balances the loss/weighting between the two objectives specifically on this subset, theoretically preserving general helpfulness while stamping out hallucinations.

**Compliance With Llm Reviewing Policy:**

Affirmed.

**Key Questions For Authors:**

1. Again, your helpfulness prompt (Appendix line 696) explicitly divorces helpfulness from factual accuracy. I know it might be a bit of a critical comment,  but given the known biases of LLM judges to reward confident formatting, how can you be sure you aren't just identifying samples where the model was highly confident but wrong? Doesn't this artificial separation create a synthetic "conflict" that wouldn't exist in human preference data?
2. If we define true helpfulness as requiring factuality (i.e., a hallucinated answer cannot be helpful), does the "small subset" of conflicting samples disappear entirely?
3. How does your method compare conceptually and empirically to recent robust preference alignment baselines like KTO or Fact-DPO, which are specifically designed to handle conflicting or factuality-dependent preference signals?
4. Could you provide an ablation study on the conflict-detection threshold? How sensitive is the final model's utility to the exact percentage of data classified as "conflicting"?

**Limitations:**

While the authors discuss standard alignment limitations, they miss the glaring vulnerabilities in their own data pipeline. Relying on an automated LLM judge to define a highly nuanced trade-off (truthfulness vs. helpfulness) introduces massive risks of reward hacking and style bias. The authors need to explicitly discuss how the potential flaws of LLM-as-a-judge impact the integrity of their conflict-identification phase.

**Strengths And Weaknesses:**

**Strengths**

* **Soundness:** The core conceptual premise, that helpfulness and truthfulness are not mutually exclusive across the *entire* dataset, is intuitively sound and well-reasoned. The mathematical formulation of dynamically weighting the loss function on the identified subset of conflicting samples seems consistent and offers a clean optimization target.
* **Presentation:** The paper is readable and logically structured. The narrative shift from observing the rarity of these objective conflicts to designing a dynamic routing mechanism to fix them is smooth, persuasive, and very easy to follow.
* **Significance:** Addressing the "alignment tax" (where models become overly cautious and refuse benign prompts) is relevant. A scalable fix to this over-refusal issue would have an immediate practical impact on post-training alignment pipelines for the broader AI community.
* **Originality:** Moving away from static, global margin adjustments to a dynamic, sample-aware routing mechanism is a clever and fresh approach. I believe It can offer a more scalable, data-centric alternative to the computationally heavy mechanistic interpretability interventions (like residual stream orthogonalization) that have been popular recently.

**Weaknesses**

* **Soundness:** This is the paper's critical failing point. The way the "conflict" is actually measured is fundamentally questionable. In the LLM-as-a-judge prompt (Appendix, I think around line 696), the rater is explicitly instructed to evaluate helpfulness "INDEPENDENT of factual mistakes." By forcing the judge to ignore factuality, you are artificially manufacturing a "conflict." If a model outputs a beautifully formatted but entirely hallucinated Python script, your judge rates it 5/5 for helpfulness. In reality, a wrong answer is inherently unhelpful. Because your entire pipeline relies on identifying this conflict, your method is likely just overfitting to the stylistic biases of the judge rather than solving a true trade-off. Furthermore, the paper lacks a rigorous ablation on the threshold used to classify a sample as "high-conflict."
* **Presentation:** While the text is generally clear, if I wanna be fair I've got to say the paper obscures the massive vulnerabilities in its data pipeline by tucking the deeply flawed helpfulness prompt into the appendix. Such a critical, structure-defining dependency needs to be addressed and justified front and center in the main text.
* **Significance:** I feel like the empirical significance is bottlenecked by weak baselines. Comparing your method solely against mostly vanilla baselines is insufficient. To prove this adaptive sample-level approach is actually significant, it must be benchmarked against contemporary, factuality-aware preference methods (like Fact-DPO) or robust methods designed to handle noisy preference signals (like KTO).
* **Originality:** While the high-level routing concept is original, the heavy reliance on a naive, off-the-shelf LLM-as-a-judge prompt to generate the foundational labels makes the actual execution feel derivative of standard synthetic data generation techniques, detracting from the novelty of the end-to-end pipeline.

---

> ### Author Rebuttal · Authors · 2026-03-31
>
> We thank the reviewer for the careful reading and helpful feedback. We address the main concerns below:
>
> **(I) W1, W2, W4, Q1, Q2, Q4 (whether conflict is artificially introduced by “independent” helpfulness evaluation prompt):**
>
> **W1 (soundness) & W4 (originality) & Q1 (confident but wrong samples):**
> * We apologize for any confusion caused by the Appendix. The key clarification is that the LLM-as-a-judge prompt is used only for evaluation (computing metrics in Sec. 5.2), not for data labeling, conflict analysis, or model training. All conflict identification and optimization are based solely on existing annotations in UltraFeedback, where helpfulness and truthfulness are provided as separate scores without instructions enforcing independence. Therefore, our method does not rely on the “independent” assumption in the evaluation prompt. The observed conflict is a property of the dataset annotations, independent of the evaluation prompt.
> * For evaluation, the “independent” phrasing is used only to reduce redundancy between helpfulness and truthfulness metrics. Importantly, our conclusions are supported by consistent improvements across both dimensions, so confidently well-formatted but incorrect outputs are penalized across all truthfulness and hallucination metrics in Tables 1 and 2.
>
> **W2 (presentation: helpfulness prompt not in the main text):**
> * We placed the evaluation prompt in the appendix since it is used only for evaluation and does not affect our pipeline (conflict analysis or model training). We will clarify this in the main text and appendix to avoid confusion.
>
> **Q2 (does conflict disappear under stricter definition):**
> * Following common practice (e.g., UltraFeedback), helpfulness and truthfulness are modeled as continuous, independently annotated 1-5 Likert scores, rather than binary variables with a subset relation (helpful $\subset$ factual). For example, a response may be more actionable but less precise (more helpful, less truthful), or factually correct but minimal or uninformative (more truthful, less helpful).
> * Even under the stricter assumption that helpfulness requires factuality, this only removes cases that are helpful but not factual (46.34% of conflicts); disagreements still remain when responses are factual but not helpful (53.66%). Thus, conflict does not disappear, but decreases from 11.09% to 5.95% after excluding helpful-but-not-factual cases.
>
> **W1 & Q4 (ablation study on conflict threshold):**
> * In our formulation, conflict detection is not a tunable threshold but is directly defined from UltraFeedback annotations as $\Delta_{help}<0$, indicating helpfulness moves opposite to truthfulness ($>0$: aligned; $=0$: no difference; $<0$: conflict). This avoids manually thresholding and filtering conflicts, while the magnitude of conflict in Eq. (9) adaptively controls regularization strength.
> * We further test stricter definitions for in Eq. (8), which manually remove a portion of conflict samples. Results show a clear drop in helpfulness, with slight decreases in truthfulness and accuracy. This indicates that stricter thresholds remove meaningful conflict cases and weaken trade-off control. This supports using $\Delta_{help}<0$ as a direct and reasonable definition. We will include this analysis in the paper.
> | |%Conflict|Help↑|Truth↑|Acc↑|Hallu↓|
> |---|---|---|---|---|---|
> |$\Delta_{help}<-2$|1.43|3.91|4.38|84.94|9.79|
> |$\Delta_{help}<-1$|5.09|3.92|4.40|84.82|9.55|
> |Ours|16.34|3.97|4.41|85.92|9.79|
>
> **(II) W3 & Q3 (baselines):**
>
> * We agree that evaluating against strong baselines is critical. Our setup includes state-of-the-art factuality-aware and multi-objective baselines. Specifically, we include FLAME (Factuality-aware Alignment) and Alignment for Honesty, which optimize factuality via preference alignment, as well as MODPO and CPO, which extend DPO to multi-objective trade-offs. All of them go beyond vanilla DPO.
> * We also thank the reviewer for suggesting additional baselines. Conceptually, KTO improves robustness to noisy signals, while Fact-DPO refines preference pairs based on correctness. We address a complementary challenge by modeling when objective conflicts arise and adaptively regulating trade-offs. We evaluate KTO and Fact-DPO on all datasets, and our method consistently outperforms both of them, demonstrating benefits of conflict-aware adaptive optimization. We will add discussion and results in the paper.
> | |Ultra| |TruthQA| |HaluEval| |
> |---|---|---|---|---|---|---|
> | |Helpful↑|Truthful↑|Helpful↑|Truthful↑|Helpful↑|Truthful↑|
> |KTO|3.86|4.40|3.92|4.28|4.66|4.77|
> |Fact-DPO|3.84|4.39|3.91|4.26|**4.70**|4.75|
> |Ours|**3.88**|**4.43**|**3.97**|**4.41**|**4.70**|**4.79**|
>
>   | |Ultra| |TruthQA| |HaluEval| |
>   |---|---|---|---|---|---|---|
>   | |Accuracy↑|Hallu↓|Accuracy↑|Hallu↓|Accuracy↑|Hallu↓|
>   |KTO|84.67|10.20|81.88|11.14|94.05|4.55|
>   |Fact-DPO|84.29|10.77|80.54|11.02|93.60|4.95|
>   |Ours|**85.91**|**9.40**|**85.92**|**9.79**|**95.20**|**3.20**|

---

> > ### Author Rebuttal · Reviewer_YBRp · 2026-04-03
> >
> > **1. Inherited Bias in the Training Data (UltraFeedback)**
> > While I appreciate the clarification that the 'independent' evaluation prompt was not used to label the training data, this simply shifts the burden of proof to the dataset itself. UltraFeedback's scores are also generated by LLMs using specific rubrics. If the original UltraFeedback annotations inherently reward structural detail or confident tone regardless of strict factuality, isn't the $\Delta_{help}<0$ conflict you are modeling still just an artificial artifact of LLM-as-a-judge bias, rather than a genuine trade-off in human preference?
> >
> > **2. Validity of the Evaluation Metric**
> > You noted that the 'independent' phrasing in the test-time helpfulness prompt was used to 'reduce redundancy' between metrics. However, if your final evaluation explicitly instructs the judge to ignore factual mistakes when scoring helpfulness, a model could theoretically output a highly detailed, beautifully formatted, but entirely hallucinated response and still receive a 5/5 for helpfulness. How can we trust the helpfulness preservation claims of a hallucination mitigation technique if the evaluation metric structurally turns a blind eye to factuality?
> >
> > **3. Annotation Noise vs. Genuine Conflict**
> > Thank you for the ablation on the conflict thresholds. However, if I'm not mistaken, relying on $\Delta_{help}<0$ assumes that any negative shift in the helpfulness score indicates a true objective conflict. On a 1-5 rating scale generated by an LLM, a difference of -1 is frequently within the standard margin of variance or noise. How does your method distinguish between a genuine semantic conflict between truthfulness and helpfulness versus standard LLM annotation noise?
> >
> > **4. Baseline Tuning Parity**
> > I appreciate the addition of the KTO and Fact-DPO baselines. Given that Fact-DPO and KTO are highly sensitive to their specific hyperparameter configurations (such as the KL penalty or the choice of the reference policy), could you confirm if these new baselines underwent the same rigorous hyperparameter tuning as CAMP, or were they run off-the-shelf?

---

> > > ### Author Response · Authors · 2026-04-07
> > >
> > > We thank the reviewer for the detailed questions.
> > >
> > > **1. Inherited Bias in Training Data**
> > > * We first clarify that we do not claim to discover objective conflicts for the first time. Prior work has widely documented trade-offs among alignment objectives, particularly in hallucination mitigation (e.g., “I don’t know” improves truthfulness but reduces helpfulness). Our contribution is to quantify the prevalence of such conflicts in representative training datasets and explicitly model them in preference optimization.
> > > * Regarding LLM-annotated data, we acknowledge potential biases. However, LLM-based annotation is now standard in large-scale alignment due to scalability, and has been shown to be a reasonable proxy for human preference. UltraFeedback is also constructed from diverse sources and models, reducing reliance on any single annotator or rubric.
> > > * Importantly, our analysis does not assume every identified conflict reflects a “pure” human preference trade-off. Instead, we study conflicts induced by widely used alignment signals that already shape modern LLM behavior through training. Even if partially influenced by annotation bias, these signals remain practically relevant for understanding and improving current alignment training.
> > >
> > > **2. Validity of Evaluation Metric**
> > > * We clarify that factuality is already explicitly evaluated through truthfulness and accuracy metrics. The helpfulness metric is therefore not intended to measure factuality again, but to capture a complementary dimension: whether a response provides useful and relevant information given the question.
> > > * This follows common practice in RLHF and alignment literature, where helpfulness, truthfulness, and harmlessness are treated as independent objectives reflecting different aspects of model behavior (e.g., modeled as orthogonal axes in CPO). For example, responses such as “I don’t know” are considered truthful but minimally helpful, illustrating the trade-off where improving truthfulness (e.g., via uncertainty or refusal) can reduce informativeness.
> > > * We understand the reviewer’s suggestion of making helpfulness dependent on factuality. However, current human preference datasets do not adopt this definition; otherwise, truthfulness would already be ensured through helpfulness training and would not remain a separate challenge, as reflected by ongoing work on factuality-aware alignment (e.g., FLAME, Fact-DPO).
> > > * To further address this concern, we additionally report a joint metric (Truthfulness $\times$ Helpfulness) that assigns high scores only when both dimensions are strong. For example, a response with (truthfulness 5, helpfulness 5) yields a score of 5, while (truthfulness 1, helpfulness 5) yields 1, ensuring that beautifully formatted but hallucinated responses are penalized (score 1). Our method continues to outperform all baselines.
> > > |Method|UltraFeedback|TruthfulQA|HaluEval|
> > > |---|---|---|---|
> > > |SFT|3.18|3.23|4.53|
> > > |DPO|3.50|3.49|4.54|
> > > |AlignHonesty|3.37|3.36|4.47|
> > > |FLAME|3.43|3.43|4.55|
> > > |MODPO|3.53|3.52|4.52|
> > > |CPO|3.47|3.34|4.53|
> > > |KTO|3.53|3.49|4.52|
> > > |F-DPO|3.51|3.46|4.55|
> > > |Ours|**3.55**|**3.60**|**4.56**|
> > >
> > > **3. Annotation Noise vs. Genuine Conflict**
> > > * We clarify that using $\Delta_{help} < 0$ does not assume every negative shift reflects a “true” semantic conflict. Similar to standard preference learning, where score differences > 0 are used to construct training signals without assuming noise-free labels, our approach operates under the presence of annotation noise. Our goal is not to explicitly separate noisy versus “true” conflict cases, but to leverage these signals as part of learning. Prior work (e.g., $\beta$-DPO) shows that small score differences often correspond to informative, fine-grained distinctions rather than noise. Therefore, including small-gap cases follows common practice in preference optimization, whereas excluding them would require stronger evidence that they are predominantly noise or even harmful to learning, which is not supported by prior findings.
> > > * Importantly, our method accounts for this by modulating optimization strength based on the magnitude of score differences. Small differences (e.g., -1) lead to weak adjustments, limiting their influence, while larger differences induce stronger updates. This naturally reduces the impact of potentially noisy signals without requiring manual filtering.
> > >
> > > **4. Baseline Tuning**
> > > * Following common practice for fair comparison, all baselines, including KTO and Fact-DPO, were implemented from official codebases and evaluated under a consistent setup (same backbone, data, and training configurations). We performed hyperparameter tuning within reasonable ranges reported in prior work to ensure competitive performance. We do not observe strong sensitivity of KTO and Fact-DPO to parameter variations within commonly used ranges. In addition, our gains are not driven by extensive tuning; instead, our method shows stable improvements across parameters and settings.

---

### Decision · Program_Chairs · 2026-04-30

**Decision:**

Accept (regular)

**Comment:**

This paper studies the trade-off between truthfulness and helpfulness in LLM alignment and proposes CAMP, a conflict-aware extension of DPO that adaptively modulates optimization at the sample level. The paper is clearly written, and reviewers agree that the observation of sparse conflicts is insightful. The method is simple, well-motivated, and shows consistent empirical improvements across benchmarks.

However, several limitations remain. The approach relies on conflict signals derived from LLM-annotated data, raising concerns about whether these reflect genuine preferences or annotation biases. In addition, the method can be viewed as an incremental extension of existing DPO-style approaches, and the evaluation relies heavily on LLM-as-a-judge metrics with limited validation across settings.

The rebuttal clarifies several points and strengthens the empirical case, though concerns about the reliability of the conflict signal and generalization remain. Overall, I find the contribution to be incremental but useful, and I lean toward a borderline accept, contingent on clearer discussion of limitations in the final version.